# Estimation of the External Knee Adduction Moment during Gait Using an Inertial Measurement Unit in Patients with Knee Osteoarthritis

**DOI:** 10.3390/s21041418

**Published:** 2021-02-18

**Authors:** Yu Iwama, Kengo Harato, Shu Kobayashi, Yasuo Niki, Naomichi Ogihara, Morio Matsumoto, Masaya Nakamura, Takeo Nagura

**Affiliations:** 1Department of Orthopaedic Surgery, Keio University School of Medicine, 35 Shinanomachi, Shinjuku-ku, Tokyo 160-8582, Japan; u-33@hotmail.co.jp (Y.I.); shu39shu@yahoo.co.jp (S.K.); y-niki@a8.keio.jp (Y.N.); morio@a5.keio.jp (M.M.); masa@keio.jp (M.N.); 2Department of Biological Sciences, The University of Tokyo, 7-3-1 Hongo, Bunkyo-ku, Tokyo 113-0032, Japan; ogihara@bs.s.u-tokyo.ac.jp; 3Department of Clinical Biomechanics, Keio University School of Medicine, 35 Shinanomachi, Shinjuku-ku, Tokyo 160-8582, Japan; nagura@keio.jp

**Keywords:** knee osteoarthritis, knee adduction moment, inertial measurement unit

## Abstract

Although the external knee adduction moment (KAM) during gait was shown to be a quantitative parameter of medial knee osteoarthritis (OA), it requires expensive equipment and a dedicated large space to measure. Therefore, it becomes a major reason to limit KAM measurement in a clinical environment. The purpose of this study was to estimate KAM using a single inertial measurement unit (IMU) during gait in patients with knee OA. A total of 22 medial knee OA patients (44 knee joints) performed conventional gait analysis using three-dimensional (3D) motion capture system. At the same time, we attached commercial IMUs to six body segments (sternum, pelvis, both thighs, and both shanks), and IMU signals during gait were recorded synchronized with the motion capture system. The peak-to-peak difference of acceleration in the lateral/medial axis immediately after heel contact was defined as the thrust acceleration (TA). We hypothesized that TA would represent the lateral thrust of the knee during the stance phase and correlate with the first peak of KAM. The relationship between the peak KAM and TA of pelvis (R = 0.52, *p* < 0.001), shanks (R = 0.57, *p* < 0.001) and thighs (R = 0.49, *p* = 0.001) showed a significant correlation. The root mean square error (RMSE) of linear regression models of pelvis, shanks, and thighs to estimate KAM were 0.082, 0.079, and 0.084 Nm/(kg·m), respectively. Our newly established parameter TA showed a moderate correlation with conventional KAM. The current study confirmed our hypothesis that a single IMU would predict conventional KAM during gait. Since KAM is known as an indicator for prognosis and severity of knee OA, this new parameter has the potential to become an accessible predictor for medial knee OA instead of KAM.

## 1. Introduction

Clinically, symptomatic knee osteoarthritis (OA) will deteriorate daily activities in the elderly population, and occur in approximately 13.6% of men and 16.5% of women over the age of 56 [1]. Therefore, it is essential to identify the disease severity in patients with knee OA. For this reason, several knee scoring systems are used to assess subjective pain, activities, and patient-reported outcome. However, dynamic function is difficult to evaluate in those patients. Thus, gait analysis system is widely used to assess the dynamic knee function, including kinematics and kinetics. In particular, the external knee adduction moment (KAM) was identified a clinical surrogate measure of medial tibiofemoral joint loading, in patients with medial knee OA [2]. Mechanical overload and abnormal kinematics affect the knee joint and can be important factors for the onset of knee OA [3]. Several studies showed that KAM during gait is associated with progression and prognosis of knee OA [2,4,5,6,7]. For instance, Hurwitz et al. reported that peak KAM was strongly correlated with the mechanical axis in knee OA patients and showed moderate correlation with OA severity [4]. Miyazaki et al. followed knee OA patients for 6 years, and showed that the group with OA progression had a significantly higher baseline KAM compared to the group without OA progression [5]. These studies showed that KAM is a reasonable parameter of knee joint load and it can be used to predict the severity and prognosis of knee OA, and thus can be used as a ‘digital biomarker’. Although KAM was shown to be a reasonable measurement of knee OA, it requires expensive equipment and a dedicated large space to measure. Therefore, it becomes a major reason to limit KAM measurement in clinical environment.

On the other hand, the Inertial Measurement Unit (IMU) is a wearable device that consists of tri-axial accelerometers and gyroscopes. In recent years, estimating kinetic gait parameters by means of wearable sensors, such as IMUs, has come to be a hot topic. Several reports on gait analysis showed the possibility of analyzing joint moments using IMU with mobile force plates. For example, Liu et al. developed a wireless sensor system consisting of a mobile force plate system, three-dimensional motion sensor units, and a wireless data logger. They applied their sensor system to calculate triaxial joint moments of the ankle, knee, and hip joints [8]. In addition, Khurelbaatar et al. indicated that both joint forces and moments in human whole-body joints using wearable inertial motion system, and in-shoe pressure sensors were feasible for normal motions with a low speed, such as walking [9]. However, these wearable systems combine multiple sensors and mobile force plates, and require complicated data processing. To our knowledge, there are no reports of measuring KAM using a single IMU that makes it possible to monitor KAM in the patient’s daily life or outpatient clinic.

The purpose of this study was to estimate KAM using a single IMU, and to evaluate the possibility for practical measurement of KAM during gait in patients with knee OA, compared to KAM obtained with conventional motion capture analysis system. It was hypothesized that a single IMU attached to the lower limb could predict conventional KAM during gait.

## 2. Materials and Methods

### 2.1. Study Design

This cross-sectional study was conducted in accordance with the Declaration of Helsinki.

### 2.2. Subjects

A total of 44 knee joints in 22 patients (19 females and 3 males) with bilateral knee OA were enrolled in the present study. Mean age was 68.5 ± 6.4 years old and mean Body Mass Index (BMI) was 22.3 ± 2.6 kg/m^2^. In this study, the radiographic findings of knee joint were evaluated on the basis of the Kellgren-Lawrence (KL) classification [10], and patients with KL grade 1 showed symptoms such as pain or stiffness in the knee joint and tenderness or crepitus at the medial joint line, and patients with KL grade 2 or higher were defined as medial knee OA [11,12]. We recruited patients who visited our outpatient department from March 2017 to April 2019, and were diagnosed with bilateral medial knee OA. All invited patients agreed to participate in the study. Patients with any symptoms in either the hip or the ankle joint were excluded from the study. Patients with any disorders that affect gait activity such as rheumatoid arthritis and lumbar spinal stenosis were also excluded. Each participant was given a written informed consent and the study protocol was approved by our ethical committee (#20150320).

### 2.3. Patient Demographics

A total of 9, 15, 13, and 7 knees were allocated to the KL grade 1, 2, 3, and 4, respectively on plain radiographs (Table 1). Mean FTA was 178.9 ± 4.0 degrees.

### 2.4. Testing Procedure

Conventional gait analysis was performed using a three-dimensional motion capture system consisting of 8 cameras (200 frames/s; Oqus, Qualisys, Sweden) and 2 force plates (frequency 2000 Hz; AM6110, Bertec, Columbus, OH, USA). A total of 46 retroreflective markers (14 mm in diameter) were attached to the standardized bony landmarks [13]. The force plate collected ground reaction force (GRF) data at 2000 Hz and were synchronized to the camera sampling rate (200 Hz). After a few practice trials, the subjects performed 6–10 trials of 10-m level walking at a self-selected speed in a gait laboratory. Among trials, the data of one successful trial without a marker drop that clearly hit the force plate was used. The motion of markers was recorded by the Qualisys Track Manager Software (version 2.7). To calculate knee kinematics and kinetics, Visual 3D (C-motion Company, Rockville, MD, USA) was used. The first peak value of KAM during the stance phase was defined as peak KAM. At the same time, we used a commercial IMU (TSND151, ATR-Promotions, Kyoto, Japan) to measure the gait characteristics. Six IMUs were attached to the body segments (sternum—the center of sternal body, pelvis—the midpoint between the posterior superior iliac spine, both thighs—anterior of the middle of femur, and both shanks—anterior of the upper third of tibia) along the direction that their Z axis of local frame was oriented toward the sagittal plane of the lab coordinate system when standing upright (Figure 1). Sensor signals during gait were recorded, synchronized with the motion capture system, using a trigger sync signal. Ranges to measure the acceleration were set to ± 4G (G = 9.81 m/s^2^). Raw sensor data were sampled at 50 Hz (13 cases) or 200 Hz (9 cases). 

In the preliminary measurement, we found that there were acceleration peaks in the lateral/medial axis, that is, the Y axis of IMU’s local frame, immediately after the heel contact identified by the synchronized data of vertical GRF. The peaks occurred simultaneously to either peaks of the knee varus angle and KAM (Figure 2). The acceleration peaks were found in all six IMUs in each gait cycle, and thus we defined the peak-to-peak difference as thrust acceleration (TA). (Figure 3)

### 2.5. Evaluations of the Clinical and Gait Parameters

As a radiographic assessment of first visit to our outpatient department, femorotibial angle (FTA) was evaluated for each patient on anteroposterior radiographs of the weight-bearing leg, on standing. The FTA was defined as the lateral angle formed by the femoral shaft and tibial shaft. Additionally, the correlation between peak KAM obtained from conventional gait analysis and TA from IMUs was examined on the sternum, pelvis, both thighs, and both shanks, respectively.

### 2.6. Statistical Analysis

Normality assumption was first performed using the Kolmogorov-Smirnov test. Thereafter, Pearson’s coefficient was used to analyze the correlation between KAM and TA. To evaluate the accuracy of a linear regression model, the root mean square error (RMSE) was used. Values of *p* < 0.05 were considered to be statistically significant. All statistical analyses were performed using SPSS^®^ version 26 (IBM, New York, NY, USA).

## 3. Results

### 3.1. Gait Parameters

The mean gait speed of all patients was 0.99 ± 0.21 m/s. The mean peak KAM was 0.33 ± 0.10 Nm/(kg·m), the mean TA of the sternum, pelvis, both thighs, and both shanks were 4571 ± 1007, 6987 ± 2790, 14,962 ± 7749, and 18,839 ± 9995 × 10^−4^ G (G = 9.81 m/s^2^), respectively.

### 3.2. Relationship of Parameters

The relationship between peak KAM and TA on the IMU of pelvis is shown in Figure 4. Each parameter showed a significant correlation (R = 0.52, *p* < 0.001). Similarly, the relationship between peak KAM and TA on the IMUs of both thighs and both shanks are shown in Figure 5 and Figure 6, respectively. Although both IMUs of shanks (R = 0.57, *p* < 0.001) and thighs (R = 0.49, *p* = 0.001) showed a significant correlation, the IMU of shanks had a higher correlation. The IMU of sternum did not show any correlation. The RMSE of linear regression models of pelvis, both shanks, and both thighs to estimate KAM were 0.082, 0.079, and 0.084 Nm/(kg·m), respectively.

## 4. Discussion

In the present study, we aimed to estimate KAM using a single IMU, and to evaluate the possibility for practical measurement of KAM during gait in patients with knee OA, compared to KAM obtained with the conventional motion capture analysis system. The present result supported the hypothesis that a single IMU would predict conventional KAM during gait. The most important finding of the present study was that TA using IMU showed a moderate correlation with conventional KAM. TA measurement was based on the peak-to-peak difference of acceleration in the lateral/medial axis occurring at the immediate after heel strike, which corresponds to the lateral thrust during the stance phase of gait. Since KAM is considered to be a quantitative measurement of lateral thrust during gait [11], our findings of moderate correlation between KAM and TA are theoretically reasonable. 

Recently, joint kinetics were measured using systems that combine wearable devices and mobile force plates [8,9]. In addition, Konrath et al. reported a method that estimates KAM and tibiofemoral joint contact force using only IMUs [14]. They previously reported an inertial motion capture system that consisted of 17 IMUs mounted on full body parts [15]. The system could build a musculoskeletal model using only inertial motion-capture-derived kinematics. They applied this system to estimate KAM and tibiofemoral joint contact force and showed comparable accuracy with RMSE between 0.006–0.014 body weight * body height and 0.4 to 1 body weights, respectively. Although KAM can be estimated by these systems, it is necessary to perform complex processing of the obtained data by combining a plurality of devices. On the other hand, our new parameter, the TA, is raw data obtained from a single IMU and thus it can be applicable in any space where the patient is able to walk for approximately 5 m. Therefore, this parameter makes it possible to estimate KAM of knee OA patients more easily, in daily life or outpatient clinics.

In terms of the degree of correlation, though the TA of pelvis, shanks, and thighs all had a moderate correlation with KAM, pelvis (R = 0.52) and shanks (R = 0.57) seemed to be better than thighs (R = 0.49) to estimate KAM. Lateral thrust as well as KAM is known to be a dynamic mechanical factor that load the medial compartment of the knee joint [16,17,18]. Previous studies already showed that OA patients with thrust show significantly higher KAM and a 4-fold increase in the odds of medial tibiofemoral OA progression [16]. The amount of lateral thrust could be measured by differences in the hip–knee–ankle angles between heel strike and the first varus peak, and was significantly correlated with the maximum value of KAM during the stance phase [17]. On the other hand, the first acceleration peak appeared in the lateral direction after the heel strike was reported as a quantitative value of lateral thrust [19]. Presumably, results of the present study indicated that the TA of thighs and shanks would correspond to the peak of knee varus angle, immediately after the heel strike and to the acceleration of the lateral thrust motion. Compared to thighs, IMUs of shanks can be mounted closer to the bone, and thus, correlation between KAM and TA on shanks seemed to be better. Furthermore, it was reported that the increased lateral thrust is already present in the early stage of OA, while increased KAM is only present in the more established phase, suggesting that increased lateral thrust is more sensitive to knee OA than KAM [18]. Considering this, TA can be a better predictor of knee OA than KAM.

The present study showed that the TAs of IMUs attached to the lower limbs and pelvis were correlated with conventional KAM during gait. This meant that it is possible to estimate and monitor KAM using only a single IMU in daily life. In the current study, the TA was also recognized at the same time as the peak of the knee varus angle. Many previous studies focused on proximal adaptation in the frontal plane for knee OA [20,21,22,23]. Harato et al. simulated knee flexion contracture and showed that the shoulder tilted to the contracture side and the pelvis tilted to the contralateral side immediately after the heel-strike of the contracture side foot [20]. That motion is consistent with the lateral trunk lean. Hunt et al. reported that lateral trunk lean explains substantial variation in the KAM in patients with medial knee OA [21]. They also showed that lateral trunk lean increases with the radiographic severity of OA [22]. Furthermore, Iijima et al. reported that an increased lateral trunk lean angle is associated with increased KAM in OA subjects [23]. Pelvic acceleration might represent the proximal adaptation to knee OA such as a lateral trunk lean. The sternum acceleration was not correlated with KAM, but this was probably due to measurement errors caused by the placement of an IMU on the sternum. We attached the IMU of sternum on a skinny shirt, not directly to the body, to avoid discomfort especially with a female subject; this setting might increase the measurement error. 

Several limitations should be noted in the current investigation. First, our newly established parameter TA could estimate KAM, but could not measure true KAM. In this study, to simplify the algorithm we only used lateral/medial acceleration to find correlation with KAM, while IMU had 6 channels of acceleration and angular velocity parameters. Other 5 parameters could be used to obtain better estimation of KAM. For this purpose, an algorithm using deep learning with more data could be more applicable. Second, the present study was limited to medial knee OA. It was reported that the lateral knee OA shows the acceleration of a medial thrust pattern, and further study is required to confirm whether the medial thrust occurs in lateral knee OA [19]. Third, we did not compare the results based on gender difference or OA severity. Previous reports said that men and women have different thresholds for radiographic severity related to knee pain [24,25]. We need to increase the number of subjects and add further consideration in the future. Fourth, we changed the measurement protocol between first 26 knees and late 18 knees, to see the influence of the sampling rate of IMU. Measurements at 200 Hz was thought to be more accurate, but the results were not significantly different between 50 Hz and 200 Hz. Lastly, the mean BMI of our patients was 22.3 kg/m^2^. Therefore, the present results might not be applicable to obese people with knee OA.

In conclusion, our newly established parameter TA showed a moderate correlation with conventional KAM. The current study confirmed our hypothesis that a single IMU would predict conventional KAM during gait. This new parameter has the potential to be a better digital biomarker for knee OA.

## Figures and Tables

**Figure 1 sensors-21-01418-f001:**
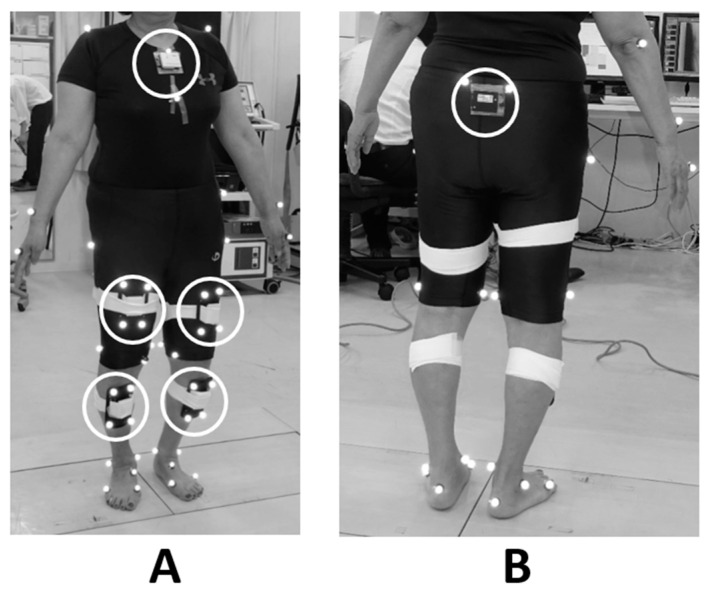
Placement of retro-reflective markers and the inertial measurement unit. (**A**) Front side and (**B**) Back side.

**Figure 2 sensors-21-01418-f002:**
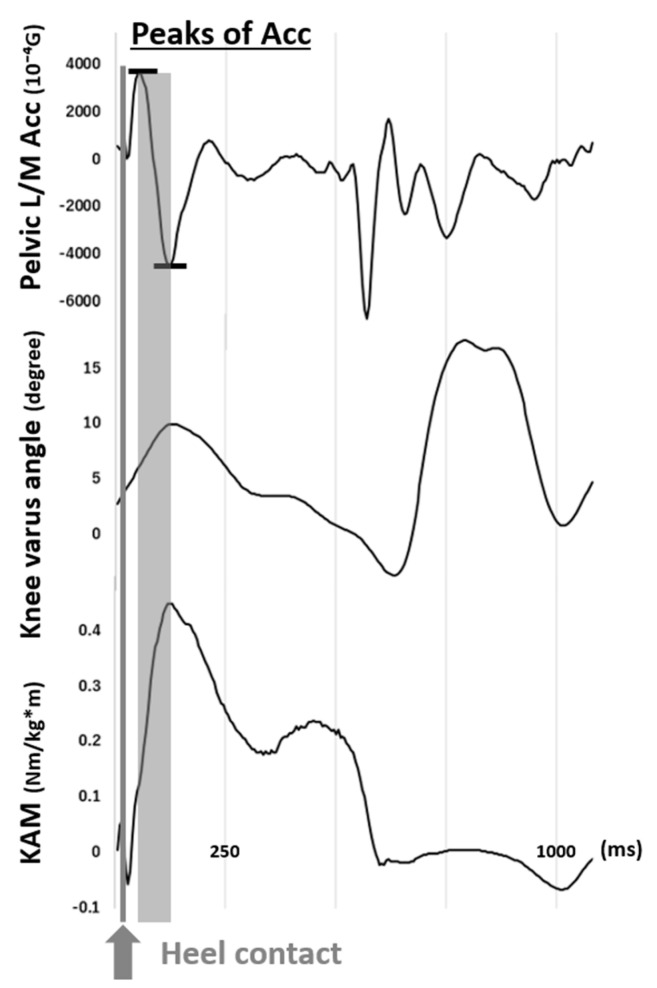
The synchronized data of the peaks of pelvic acceleration (Acc) in the lateral/medial (L/M) axis, knee varus angle, and knee adduction moment (KAM) from a representative case. (G = 9.81 m/s^2^).

**Figure 3 sensors-21-01418-f003:**
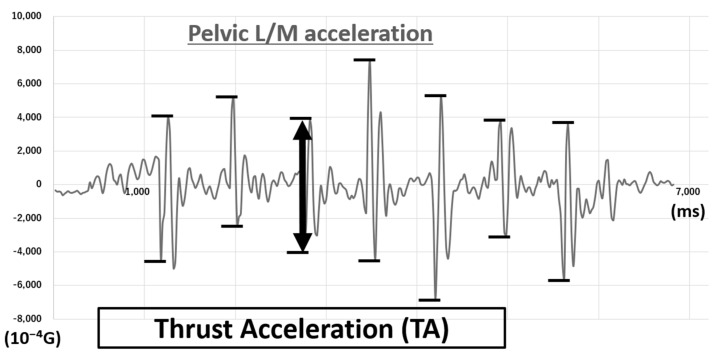
The peak-to-peak difference of acceleration in the lateral/medial (L/M) axis using a pelvic inertial measurement unit (IMU) from a representative case. (G = 9.81 m/s^2^).

**Figure 4 sensors-21-01418-f004:**
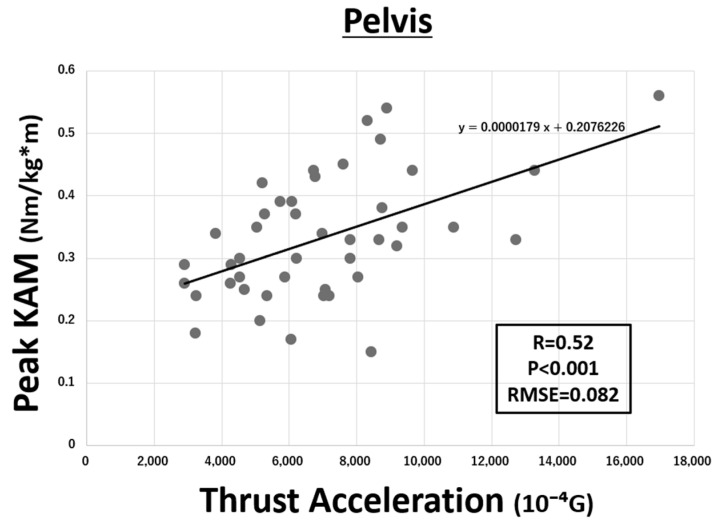
Correlation between the peak knee adduction moment (KAM) and thrust acceleration (TA) on the pelvis (R = 0.52, *p* < 0.001). The root mean square error (RMSE) was 0.082 Nm/(kg·m) (G = 9.81 m/s^2^).

**Figure 5 sensors-21-01418-f005:**
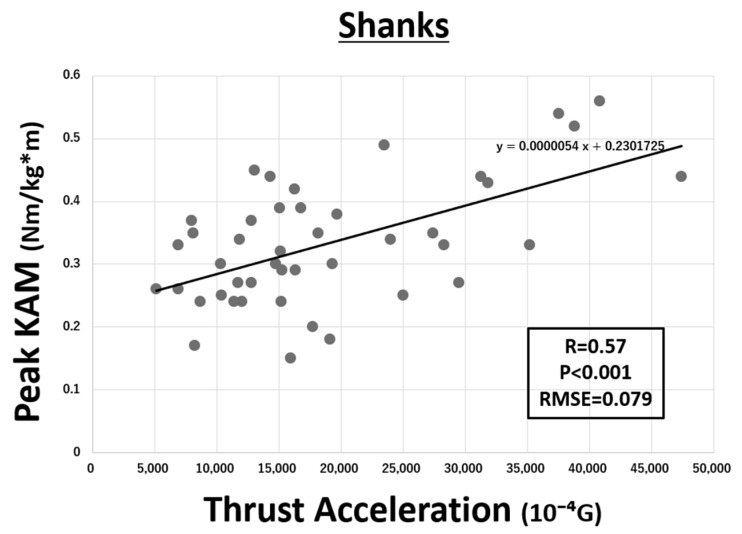
Correlation between peak knee adduction moment (KAM) and thrust acceleration (TA) on both shanks (R = 0.57, *p* < 0.001). The root mean square error (RMSE) was 0.079 Nm/(kg·m) (G = 9.81 m/s^2^).

**Figure 6 sensors-21-01418-f006:**
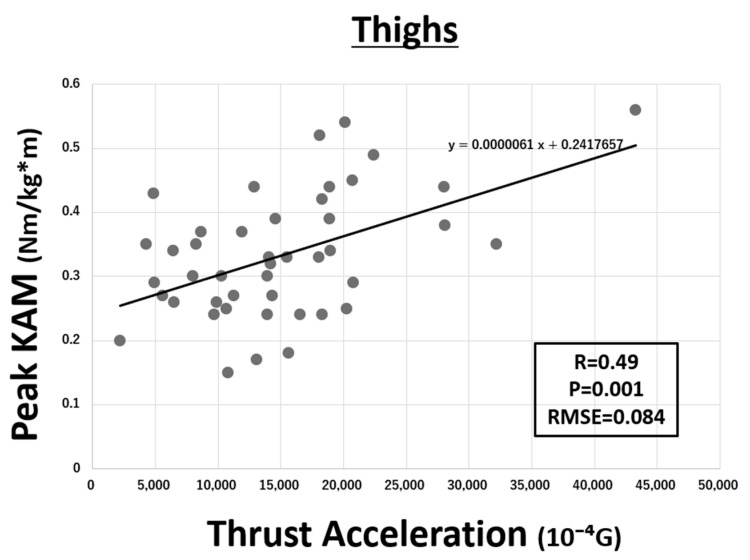
Correlation between peak knee adduction moment (KAM) and thrust acceleration (TA) on both thighs (R = 0.49, *p* = 0.001). The root mean square error (RMSE) was 0.084 Nm/(kg·m) (G = 9.81 m/s^2^).

**Table 1 sensors-21-01418-t001:** Patient demographics (mean ± SD).

Age (Years)	68.5 ± 6.4
Gender (Female/Male)	19/3
Body Mass Index (kg/m^2^)	22.3 ± 2.6
Femoro-Tibial Angle (degree)	178.9 ± 4.0
Kellgren-Lawrence grade (1/2/3/4)	9/15/13/7

## Data Availability

The data presented in this study are available on request from the corresponding author. The data are not publicly available due to privacy.

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
