# Peer review of "Estimation of the External Knee Adduction Moment during Gait Using an Inertial Measurement Unit in Patients with Knee Osteoarthritis"

_sensors, 2021, doi:10.3390/s21041418_

Round 1
Reviewer 1 Report
This manuscript describes an effort to fit a linear regression model in order to predict the peak Knee Adduction Moment (KAM), which is cumbersome to measure, from measurements of medio-lateral (sideway) acceleration with one (or a few) IMUs, which is a much easier experiment.
The study is well motivated, but I find the manuscript lacking in the description of some important parts. I am also sceptical about the usefulness of the result considering the only moderate correlation obtained with the suggested model.
Major comments:
- It appears plausible that there should be a (causal) connection between the medio-lateral acceleration of the shank, thigh and pelvis, and the adduction moment of the knee. However, the correlation observed (r between 0.49 and 0.57) is moderate at best. Is this level of correlation in a regression model really useful in a clinical situation? Calculating the RMSE should be done (and reported in the article) which would give a value for the typical error in making a prediction using the fitted regression model. From figure 5 it seems that RMSE could be of the order 1e4 Nm/(kg*m). Is this level of error acceptable? These are important points for discussion.
- There are important details missing regarding the actual placement of the IMUs. This is very important for repeatability (reliability) of the method.
- It is not clear to me how the actual signal of the L/M acceleration was obtained, but I assume it is the raw accelerometer signal of one of the channels from the IMU. This makes the method very sensitive to accurate placement and orientation of the sensor. An anatomically relevant frame of reference could be obtained by estimating the knee joint axis from the sensor data. See, for instance, Olsson et al. However, this implies two IMUs, one on the thigh and the other on the shank. - Minor comments
- Line 51: "... disputed ... " unsuitable choice of word
- Line 70: "Kellgren-Lawrence classification", please provide a reference
- Line 85: More precise definition of IMU placement, please.
- Line 90: How was ML axis determined for IMUs?
- Line 92: "... range of the peaks ..." better with "peak-to-peak difference"
- 3.1 Patient demographics, belongs in Methods section not Results, since it is not an outcome. The femorial-tibial angle could also be regarded as describing the sample population, and not a result.
- Lines 157-168: This paragraph describes the importance of measuring/estimating KAM, but do not discuss results obtained. The paragraph belongs in the Introduction.
- Line 174: "... comparable accuracy" is a very weak statement. Please be more precise using numerical values for the accuracy in the comparison.
- Lines 180-181: what does "dynamic mechanical factor" mean in this context?
- Lines 183-184: "... difference in the hip knee angles ..." Which angles are meant must be more precisely defined in 3D human motion.
- Lines 214: Not sure if deep learning will be useful to find a regression model without collecting a lot more data.
Olsson, F., Kok, M., Seel, T., & Halvorsen, K. (2020). Robust plug-and-play joint axis estimation using inertial sensors. Sensors, 20(12), 3534.
Author Response
Manuscript ID: sensors-1100814
Type: Communication
Title: Estimation of the external knee adduction moment during gait using an inertial measurement unit in patients with knee osteoarthritis
Authors: Yu Iwama , Kengo Harato * , Shu Kobayashi , Yasuo Niki , Naomichi Ogihara , Morio Matsumoto , Masaya Nakamura , Takeo Nagura
Response to Reviewers
Reviewer 1
Major comments:
It appears plausible that there should be a (causal) connection between the medio-lateral acceleration of the shank, thigh and pelvis, and the adduction moment of the knee. However, the correlation observed (r between 0.49 and 0.57) is moderate at best. Is this level of correlation in a regression model really useful in a clinical situation? Calculating the RMSE should be done (and reported in the article) which would give a value for the typical error in making a prediction using the fitted regression model. From figure 5 it seems that RMSE could be of the order 1e4 Nm/(kg*m). Is this level of error acceptable? These are important points for discussion.
Author Response
First of all, we appreciate the comments from the Reviewer. We calculated the RMSE of a linear regression model and found it to be 0.082, 0.079 and 0.084 Nm/kg*m for the pelvis, shanks and thighs respectively. Although the accuracy of these models is not high, TA is a parameter that can be easily obtained from raw data of a single IMU, so it has sufficient clinical value to be able to easily estimate KAM in daily life. The manuscript was revised as follows:
(page 1, line 29-31) The root mean square error (RMSE) of linear regression models of pelvis, shanks and thighs to estimate KAM were 0.082, 0.079 and 0.084 Nm/kg*m, respectively.
(page 5, line 160) To evaluate the accuracy of a linear regression model, the root mean square error (RMSE) was used.
(page 6, line 181-182) The RMSE of linear regression models of pelvis, both shanks and both thighs to estimate KAM were 0.082, 0.079 and 0.084 Nm/kg*m, respectively.
- There are important details missing regarding the actual placement of the IMUs. This is very important for repeatability (reliability) of the method.
Author Response
Thank you very much for the comments from the Reviewer. We added the statement concerning the detailed placement of the IMUs. The manuscript was revised as follows:
(page 3, line 120-124) Six IMUs were attached to body segments (sternum: the center of sternal body, pelvis: the midpoint between the posterior superior iliac spine, both thighs: anterior of the middle of femur, and both shanks: anterior of the upper third of tibia) along the direction that their Z axis of local frame was oriented toward the sagittal plane of the lab coordinate system when standing upright (Figure 1).
- It is not clear to me how the actual signal of the L/M acceleration was obtained, but I assume it is the raw accelerometer signal of one of the channels from the IMU. This makes the method very sensitive to accurate placement and orientation of the sensor. An anatomically relevant frame of reference could be obtained by estimating the knee joint axis from the sensor data. See, for instance, Olsson et al. However, this implies two IMUs, one on the thigh and the other on the shank.
Author Response
Thank you very much for the comments from the Reviewer, again. The axis of the acceleration is very important. Although the IMU coordinate system could be calibrated using multiple IMUs, considering its clinical application, we focused on easily estimating KAM from the data of a single IMU. Therefore, we attached the IMUs along the direction that their Z axis of local frame was oriented toward the sagittal plane of the lab coordinate system, and regarded the acceleration of Y axis in the local frame as the L/M acceleration and used it as a parameter. In addition to the response above, the manuscript was revised as follows:
(page 3, line 128-130) In the preliminary measurement, we have found that there were acceleration peaks in the lateral/medial axis, that is the Y axis of IMU’s local frame, immediately after heel contact identified by synchronized data of vertical GRF.
Minor comments
- Line 51: "... disputed ... " unsuitable choice of word
Author Response
Thank you for the comment. We changed the word. The manuscript was revised as follows:
(page 2, line 64-65) In the recent years, estimating kinetic gait parameters by means of wearable sensors, such as IMUs, has come to be a hot topic.
- Line 70: "Kellgren-Lawrence classification", please provide a reference
Author Response
Thank you for the comment. We provided a reference. The manuscript was revised as follows:
(page 3, line 88-89) In this study, radiographic findings of knee joint were evaluated based on the Kellgren-Lawrence (KL) classification [11]
- Line 85: More precise definition of IMU placement, please.
- Line 90: How was ML axis determined for IMUs?
Author Response
Thank you for the comments. Please see the response to major comments.
- Line 92: "... range of the peaks ..." better with "peak-to-peak difference"
Author Response
Thank you for the comment. We changed the phrase. The manuscript was revised as follows:
(page 1, line 25-27) The peak-to-peak difference of acceleration in the lateral/medial axis immediately after heel contact was defined as the thrust acceleration (TA).
(page 3, line 131-132) The acceleration peaks were found in all six IMUs in each gait cycle, and thus we defined the peak-to-peak difference as thrust acceleration (TA).
- 3.1 Patient demographics, belongs in Methods section not Results, since it is not an outcome. The femorial-tibial angle could also be regarded as describing the sample population, and not a result.
Author Response
Thank you for the comment. We moved patient demographics to Materials and Methods. The manuscript was revised as follows:
(page 3, line 100-102) 2.3. Patient demographics
A total of 9, 15, 13, and 7 knees were allocated to the KL grade 1, 2, 3, and 4, respectively on plain radiographs (Table 1). Mean FTA was 178.9 ± 4.0 degrees.
- Lines 157-168: This paragraph describes the importance of measuring/estimating KAM, but do not discuss results obtained. The paragraph belongs in the Introduction.
Author Response
Thank you for the comment. We moved the paragraph to Introduction. The manuscript was revised as follows:
(page 2, line 47-58) Mechanical overload and abnormal kinematics affect the knee joint and can be important factors for the onset of knee OA [3]. Several studies have shown that KAM during gait is associated with progression and prognosis of knee OA [2, 4-6]. For instance, Hurwitz et al. reported that peak KAM was strongly correlated with mechanical axis in knee OA patients and showed moderate correlation with OA severity [4]. Miyazaki et al. followed knee OA patients for 6 years, and showed that the group with OA progression had a significantly higher baseline KAM compared to the group without OA progression [5]. These studies show that KAM is a reasonable parameter of knee joint load and it can be used to predict the severity and prognosis of knee OA, and thus can be used as ‘digital biomarker’. Although KAM has been shown as such a reasonable measurement of knee OA, it requires expensive equipment and dedicated large space to measure. Therefore it becomes a major reason to limit KAM measurement in clinical environment.
- Line 174: "... comparable accuracy" is a very weak statement. Please be more precise using numerical values for the accuracy in the comparison.
Author Response
Thank you for the comment. We added the numerical values for the accuracy to the sentence. The manuscript was revised as follows:
(page 9, line 229-231) They applied this system to estimate KAM and tibiofemoral joint contact force and showed comparable accuracy with RMSE between 0.006–0.014 body weight * body height and 0.4 to 1 body weights, respectively.
- Lines 180-181: what does "dynamic mechanical factor" mean in this context?
Author Response
Thank you for the comment. The dynamic mechanical factor meant that lateral thrust as well as KAM is a mechanical factor that load the medial compartment of the knee joint. The manuscript was revised as follows:
(page 9, line 238-239) Lateral thrust as well as KAM has been known as a dynamic mechanical factor that load the medial compartment of the knee joint.
- Lines 183-184: "... difference in the hip knee angles ..." Which angles are meant must be more precisely defined in 3D human motion.
Author Response
Thank you for the comment. We should have described it as hip-knee-ankle angle, that is the angle between the mechanical axes of the femur and the tibia. The manuscript was revised as follows:
(page 9, line 241-244) The amount of lateral thrust can be measured by differences in the hip-knee-ankle angles between heel strike and the first varus peak, and is significantly correlated with the maximum value of KAM during stance phase [7].
- Lines 214: Not sure if deep learning will be useful to find a regression model without collecting a lot more data.
Author Response
Thank you for the comment. We think it is necessary to collect quite a lot of data to use deep learning to improve the accuracy of the regression model. The manuscript was revised as follows:
(page 10, line 273-274) For this purpose, an algorithm using deep learning with more data could be applicable.

Reviewer 2 Report
BRIEF SUMMARY
The study aimed to establish correlation between knee adduction moment (KAM) derived from a laboratory-based motion capture system, and a thrust acceleration (TA), defined as range of peak-to-peak acceleration in the lateral/medial axis and derived from inertial measurement units (IMU) placed on multiple body segments. Authors recruited 22 people with knee osteoarthritis (OA) and observed significant correlation between the peak KAM and TA of pelvis, shanks and thighs, but not TA of sternum.
BROAD (MAJOR) COMMENTS:
I congratulate authors on their work. This is a well-written paper with informative figures. Overall, I found the topic timely and clinically important. This paper contributes to the increasing trend of assessing human motion outside laboratory and provides helpful information regarding obtaining KAM using wearable sensors. My main concern is:
Poor reporting. Please format the paper according to reporting guidelines such as CONSORT or STROBE. Although it might not be directly applicable to this study design, it will greatly facilitate reading. Information on study design, setting, recruitment procedures, definition of outcomes, data postprocessing is currently lacking or superficially touched upon, and should be provided in separate paragraphs to enhance reading and quality of the paper. Details below.
SPECIFIC COMMENTS
ABSTRACT
Line 18: “good quantitative parameter of medial knee OA”. As far as I am concerned, the literature is rather conflicting on this topic. Please rephrase to better represent link between KAM and knee OA.
Line 33: “has the potential to become simple and reasonable”. It might be simpler but not necessarily simple as it still requires significant amount of data processing (identification of heel strikes from acc signal etc). Also, not sure what do you mean by reasonable in this context. Please rephrase.
INTRODUCTION
Line 60: Please provide example of advantages, for research or clinical practise, that identification of the KAM from a single IMU will bring.
METHODS
Please start off with the paragraph entitled Study design, and carefully describe.
Was this study conducted according to the Declaration of Helsinki?
Point 2.1 Subjects:
- Please move the information regarding the demographics into the results section.
- Please provide information how did you define and measure varus deformities
- Radiographic knee OA is defined as K& L score > 2. Please correct.
- More details are needed regarding recruitment procedures. From where did you recruit participants, in what time frame and what was the recruitment rate that how many invited and how many rejected the invite or were not eligible etc.
Please provide information on the outcomes (KAM and TA) is separate points and carefully discuss.
How did you identify heel strikes from acceleration data? Did you use any specific threshold of the acceleration signal? how did you synchronize data from mocap and IMU data?
What was the length of the walkway? How many walking trials did they perform? From how many steps you collected and quantified outcomes?
Line 88: “Raw sensor data were sampled at 50Hz (13 cases) or 200Hz (9 cases)”. Please provide reason for that and whether it had any impact on the results?
Figure 3: is this from one subject or all subjects?
Line 108: “As a preoperative radiographic assessment” This would indicate that these subjects were scheduled for surgery. If so, please add this information in the relevant section.
Line 118: Have you checked data normality and did you use appropriate test based on the normality of data?
RESULTS
Have you collected data on knee pain? If so, please add. Please add average speed as well.
Line 127: “was” please correct to is
Line 129: “were” please correct to are
DISCUSSION
I suggest to start off the discussion with reminding the reader about the study objectives.
Line 157: “join”: typo, please correct
Please provide infomation how it will impact research and/or clinical practise.
You study population had BMI of 22. Please add in the limitation that your results might not be applicable to obese people with knee OA.
Author Response
Manuscript ID: sensors-1100814
Type: Communication
Title: Estimation of the external knee adduction moment during gait using an inertial measurement unit in patients with knee osteoarthritis
Authors: Yu Iwama , Kengo Harato * , Shu Kobayashi , Yasuo Niki , Naomichi Ogihara , Morio Matsumoto , Masaya Nakamura , Takeo Nagura
Response to Reviewers
Reviewer 2
BROAD (MAJOR) COMMENTS:
I congratulate authors on their work. This is a well-written paper with informative figures. Overall, I found the topic timely and clinically important. This paper contributes to the increasing trend of assessing human motion outside laboratory and provides helpful information regarding obtaining KAM using wearable sensors. My main concern is:
Author Response
First of all, we appreciate the comments from the Reviewer. Thank you very much.
Poor reporting. Please format the paper according to reporting guidelines such as CONSORT or STROBE. Although it might not be directly applicable to this study design, it will greatly facilitate reading. Information on study design, setting, recruitment procedures, definition of outcomes, data postprocessing is currently lacking or superficially touched upon, and should be provided in separate paragraphs to enhance reading and quality of the paper. Details below.
Author Response
Thank you very much for the comment. We revised the manuscript to improve the quality of reporting as much as possible. Please see the revised manuscript.
SPECIFIC COMMENTS
ABSTRACT
Line 18: “good quantitative parameter of medial knee OA”. As far as I am concerned, the literature is rather conflicting on this topic. Please rephrase to better represent link between KAM and knee OA.
Author Response
Thank you for the comment. We changed the phrase. The manuscript was revised as follows:
(page 1, line 17-18) Although the external knee adduction moment (KAM) during gait has been shown as one of quantitative parameters of medial knee osteoarthritis (OA),
(page 2, line 45-46) In particular, the external knee adduction moment (KAM) has been identified as one of clinical surrogate measures of medial tibiofemoral joint loading in patients with medial knee OA [2].
Line 33: “has the potential to become simple and reasonable”. It might be simpler but not necessarily simple as it still requires significant amount of data processing (identification of heel strikes from acc signal etc). Also, not sure what do you mean by reasonable in this context. Please rephrase.
Author Response
Thank you for the comment. We think that TA might not be simple but much easier parameter to access than KAM. The manuscript was revised as follows:
(page 1, line 33-35) Since KAM is known as an indicator for prognosis and severity of knee OA, this new parameter has the potential to become accessible predictor for medial knee OA instead of KAM.
INTRODUCTION
Line 60: Please provide example of advantages, for research or clinical practise, that identification of the KAM from a single IMU will bring.
Author Response
Thank you for the comment. The advantage of estimating KAM using a single IMU is that make it possible to monitor KAM in the patient's daily life or outpatient clinic. The manuscript was revised as follows:
(page 2, line 75-77) To our knowledge, there have been no reports of measuring KAM using a single IMU that make it possible to monitor KAM in the patient's daily life or outpatient clinic.
METHODS
Please start off with the paragraph entitled Study design, and carefully describe.
Was this study conducted according to the Declaration of Helsinki?
Author Response
Thank you for the comment. The present study is cross-sectional study conducted in accordance with the Declaration of Helsinki. The manuscript was revised as follows:
(page 2, line 84) This cross-sectional study was conducted in accordance with the Declaration of Helsinki.
Point 2.1 Subjects:
Please move the information regarding the demographics into the results section.
Author Response
Thank you for the comment. This comment conflicted with another reviewer's comment. We finally think that patient demographics could be regarded as describing the sample population and belong in Methods section.
Please provide information how did you define and measure varus deformities
Radiographic knee OA is defined as K& L score > 2. Please correct.
Author Response
Thank you for the comment. In this study, we defined medial knee OA patients as patients with KL grade 1 having symptoms such as pain or stiffness in the knee joint and tenderness or crepitus at the medial joint line, or patients with KL grade 2 or higher. The manuscript was revised as follows:
(page 2, line 88-91) In this study, radiographic findings of knee joint were evaluated based on the Kellgren-Lawrence (KL) classification [11], and patients with KL grade 1 having symptoms such as pain or stiffness in the knee joint and tenderness or crepitus at the medial joint line, or patients with KL grade 2 or higher were defined as medial knee OA [12, 13].
More details are needed regarding recruitment procedures. From where did you recruit participants, in what time frame and what was the recruitment rate that how many invited and how many rejected the invite or were not eligible etc.
Author Response
Thank you for the comment. We added the statement concerning the detailed information of recruitment procedures. The manuscript was revised as follows:
(page 2-3, line 92-95) We recruited patients who visited our outpatient department from March 2017 to April 2019, and were diagnosed with bilateral medial knee OA. All invited patients agreed with the participation of the study.
Please provide information on the outcomes (KAM and TA) is separate points and carefully discuss.
Author Response
Thank you for the comment. We provided information on the outcomes of KAM and TA respectively. The manuscript was revised as follows:
(page 6, line 172-174) The mean peak KAM was 0.33 ± 0.10 Nm/kg*m, the mean TA of sternum, pelvis, both thighs and both shanks were 4571 ± 1007, 6987 ± 2790, 14962 ± 7749 and 18839 ± 9995 10-4G (G=9.81 m/s2), respectively.
How did you identify heel strikes from acceleration data? Did you use any specific threshold of the acceleration signal? how did you synchronize data from mocap and IMU data?
Author Response
Thank you for the comment. We used a trigger sync signal to synchronize motion capture system with IMUs. In the present study, we identified heel contact by synchronized data of vertical GRF. In the process of IMU data analysis, we found that the onset of peak of vertical acceleration, that is X-axis acceleration in the local frame of IMU, tend to be coincided with the heel contact identified by the vertical GRF. By proceeding with the verification, we believe that future study can provide a protocol to identify heel contact from vertical acceleration. The manuscript was revised as follows:
(page 3, line 124-125) Sensor signals during gait were recorded synchronized with motion capture system using a trigger sync signal.
(page 3, line 128-130) In the preliminary measurement, we have found that there were acceleration peaks in the lateral/medial axis, that is the Y axis of IMU’s local frame, immediately after heel contact identified by synchronized data of vertical GRF.
What was the length of the walkway? How many walking trials did they perform? From how many steps you collected and quantified outcomes?
Author Response
Thank you for the comment. We added the statement concerning the detailed information of gait trials. The manuscript was revised as follows:
(page 3, line 114-116) After a few practice trials, the subjects performed 6-10 trials of 10m level walking at a self-selected speed in a gait laboratory. Among trials, the data of one successful trial without marker drop and clearly hit the force plate was used.
Line 88: “Raw sensor data were sampled at 50Hz (13 cases) or 200Hz (9 cases)”. Please provide reason for that and whether it had any impact on the results?
Author Response
Thank you for the comment. We changed the measurement protocol between first 26 knees and late 18 knees, to see the influence of the sampling rate of IMU. The results for each sampling rate are shown below.
Measurements at 200 Hz was thought to be more accurate, but the results were not significantly different between 50Hz and 200Hz. The manuscript was revised as follows:
(page 10, line 279-282) Fourth, we changed the measurement protocol between first 26 knees and late 18 knees, to see the influence of the sampling rate of IMU. Measurements at 200 Hz was thought to be more accurate, but the results were not significantly different between 50Hz and 200Hz.
Figure 3: is this from one subject or all subjects?
Author Response
Thank for the comment. Figure 3 is from a representative case. The manuscript was revised as follows:
(page 5, line 146-147) Figure 3. The peak-to-peak difference of acceleration in the lateral/medial (L/M) axis using a pelvic inertial measurement unit (IMU) from a representative case.
Line 108: “As a preoperative radiographic assessment” This would indicate that these subjects were scheduled for surgery. If so, please add this information in the relevant section.
Author Response
Thank you for the comment. We are sorry, preoperative was incorrect. We assessed the radiographic findings of first visit to our outpatient department. The manuscript was revised as follows:
(page 5, line 150) As a radiographic assessment of first visit to our outpatient department,
Line 118: Have you checked data normality and did you use appropriate test based on the normality of data?
Author Response
Thank you for the comment. We have checked data normality using the Kolmogorov-Smirnov test. Thereafter, Pearson’s coefficient was used to analyse the correlation between KAM and TA. The manuscript was revised as follows:
(page 5, line 158-159) Normality assumption was first performed using the Kolmogorov-Smirnov test. Thereafter, Pearson’s coefficient was used to analyse the correlation between KAM and TA.
RESULTS
Have you collected data on knee pain? If so, please add. Please add average speed as well.
Author Response
Thank you for the comment. Unfortunately, we have not collected data on knee pain. We added the average gait speed. The manuscript was revised as follows:
(page 6, line 172) The mean gait speed of all patients was 0.99 ± 0.21 m/s.
Line 127: “was” please correct to is
Author Response
Thank you for the comment. The manuscript was revised as follows:
(page 6, line 176) The relationship between peak KAM and TA on IMU of pelvis is shown in Figure 4.
Line 129: “were” please correct to are
Author Response
Thank you for the comment. The manuscript was revised as follows:
(page 6, line 177-179) Similarly, the relationship between peak KAM and TA on IMUs of both thighs and both shanks are shown in Figure 5 and Figure 6, respectively.
DISCUSSION
I suggest to start off the discussion with reminding the reader about the study objectives.
Author Response
Thank you for the suggestion. The manuscript was revised as follows:
(page 8, line 204-206) In the present study, we aimed to estimate KAM using a single IMU, and to evaluate the possibility for practical measurement of KAM during gait in patients with knee OA, compared to KAM obtained with conventional motion capture analysis system.
Line 157: “join”: typo, please correct
Author Response
Thank you for the comment. The manuscript was revised as follows:
(page 2, line 47-48) Mechanical overload and abnormal kinematics affect the knee joint and can be important factors for the onset of knee OA [3].
Please provide infomation how it will impact research and/or clinical practise.
Author Response
Thank you for the comment. TA is an accessible parameter from a single IMU. Therefore, it makes it possible to measure KAM of knee OA patients more easily in daily life or outpatient clinics. The manuscript was revised as follows:
(page 9, line 235-236) Therefore, this parameter makes it possible to measure KAM of knee OA patients more easily in daily life or outpatient clinics.
You study population had BMI of 22. Please add in the limitation that your results might not be applicable to obese people with knee OA.
Author Response
Thank you for the comment. We added the limitation concerning the obese people with knee OA. The manuscript was revised as follows:
(page 10, line 282-283) Lastly, the mean BMI of our patients was 22.3 kg/m2. Therefore, the present results might not be applicable to obese people with knee OA.

Round 2
Reviewer 1 Report
The authors have done a good job with the revision of this manuscript and addressed well my concerns from the first version.
I think the authors can still be a bit more cautious in their discussion regarding the clinical usefulness of the moderate correlations they report.
I am also curious about the unit of acceleration used in figures 4-6. Why not just report in g?
Author Response
Response to Reviewers
Reviewer 1
Comments and Suggestions for Authors
The authors have done a good job with the revision of this manuscript and addressed well my concerns from the first version.
Author Response
First of all, we appreciate the comments from the Reviewer. The comments were very helpful to us when revising the manuscript.
I think the authors can still be a bit more cautious in their discussion regarding the clinical usefulness of the moderate correlations they report.
Author Response
Thank you very much for the comment from the Reviewer. We revised the statement regarding the degree of correlation and estimating KAM. The manuscript was revised as follows:
(page 1, line 31-32) Our newly established parameter TA showed a moderate correlation with conventional KAM.
(page 9, line 208-210) The most important finding of the present study was that TA using IMU showed a moderate correlation with conventional KAM.
(page 9, line 213-214) Since KAM is considered as quantitative measurement of lateral thrust during gait [12], our findings of moderate correlation between KAM and TA are theoretically reasonable.
(page 10, line 237-238) Therefore, this parameter makes it possible to estimate KAM of knee OA patients more easily in daily life or outpatient clinics.
(page 10, line 239-241) In terms of the degree of correlation, though the TA of pelvis, shanks, and thighs all had a moderate correlation with KAM, the pelvis (R=0.52) and shanks (R=0.57) seem to be better than thighs (R=0.49) to estimate KAM.
(page 11, line 287-288) As conclusion, our newly established parameter TA showed a moderate correlation with conventional KAM.
I am also curious about the unit of acceleration used in figures 4-6. Why not just report in g?
Author Response
Thank you for the comment. We corrected the unit of acceleration for all figures to G. The manuscript was revised as follows:
(page 4, line 138-142) Figure 2. The synchronized data of the peaks of pelvic acceleration (Acc) in the lateral/medial (L/M) axis, knee varus angle and knee adduction moment (KAM) from a representative case. (G=9.81 m/s2)
(page 5, line 144-148) Figure 3. The peak-to-peak difference of acceleration in the lateral/medial (L/M) axis using a pelvic inertial measurement unit (IMU) from a representative case. (G=9.81 m/s2)
(page 7, line 186-190) Figure 4. Correlation between peak knee adduction moment (KAM) and thrust acceleration (TA) on pelvis (R=0.52, P<0.001). The root mean square error (RMSE) is 0.082 Nm/kg*m. (G=9.81 m/s2)
(page 8, line 192-196) Figure 5. Correlation between peak knee adduction moment (KAM) and thrust acceleration (TA) on both shanks (R=0.57, P<0.001). The root mean square error (RMSE) is 0.079 Nm/kg*m. (G=9.81 m/s2)
(page 9, line 198-202) Figure 6. Correlation between peak knee adduction moment (KAM) and thrust acceleration (TA) on both thighs (R=0.49, P=0.001). The root mean square error (RMSE) is 0.084 Nm/kg*m. (G=9.81 m/s2)
Reviewer 2 Report
Authors adressed all of my comments appropriately.
Author Response
Response to Reviewers
Reviewer 2
Comments and Suggestions for Authors
Authors adressed all of my comments appropriately.
Author Response
Thank you very much for the comment from the Reviewer. The comments were very helpful to us when revising the manuscript.